# A Novel Framework for Open-Set Authentication of Internet of Things Using Limited Devices

**DOI:** 10.3390/s22072662

**Published:** 2022-03-30

**Authors:** Keju Huang, Junan Yang, Pengjiang Hu, Hui Liu

**Affiliations:** College of Electronic Engineering, National University of Defense Technology, Hefei 230037, China; yangjunan@ustc.edu (J.Y.); pjhu12@nudt.edu.cn (P.H.); liuhui17c@nudt.edu.cn (H.L.)

**Keywords:** Internet of Things, cybersecurity, physical layer identification, deep learning, open-set classification

## Abstract

The Internet of Things (IoT) is promising to transform a wide range of fields. However, the open nature of IoT makes it exposed to cybersecurity threats, among which identity spoofing is a typical example. Physical layer authentication, which identifies IoT devices based on the physical layer characteristics of signals, serves as an effective way to counteract identity spoofing. In this paper, we propose a deep learning-based framework for the open-set authentication of IoT devices. Specifically, additive angular margin softmax (AAMSoftmax) was utilized to enhance the discriminability of learned features and a modified OpenMAX classifier was employed to adaptively identify authorized devices and distinguish unauthorized ones. The experimental results for both simulated data and real ADS–B (Automatic Dependent Surveillance–Broadcast) data indicate that our framework achieved superior performance compared to current approaches, especially when the number of devices used for training is limited.

## 1. Introduction

The Internet of Things (IoT), which enables communication and interaction between various devices, promises to transform a wide range of fields. IoT devices primarily transmit information via wireless means, the open nature of which exposes the IoT to cybersecurity threats [1]. One typical cybersecurity threat, identity spoofing, which refers to the action of assuming the identity of some other device, decreases the availability of resources and can be dangerous in critical infrastructures [2]. By using spoofing identities, attackers can gain unauthorized access to internal networks and interfere with communication between authorized devices, which threatens the security of the wireless network. Therefore, the network administrator must identify authorized IoT devices and reject connections from unauthorized devices (Figure 1).

To prevent identity proofing, physical layer authentication (PLA) [3], which is also known as non-cryptographic device identification (NDI) [4], identifies IoT devices based on the physical layer characteristics of their transmitted signals. The feasibility of PLA is based on the fact that the electronic circuits of devices possess specific imperfections that are determined by production and manufacturing processes. PLA is analogous to speaker recognition [5] in the sense that they both concern the characteristics of components that emit signals, regardless of the content of the signals. PLA serves as an effective tool against identity spoofing as it identifies devices using the physical layer characteristics of signals that stem from hardware imperfections, which cannot be counterfeited, in theory. Compared to cryptographic approaches for authentication, PLA does not require sophisticated key management procedures and is hard to deceive. Therefore, PLA has received a lot of attention in the past few years.

Existing approaches to PLA can be divided into two categories: hand-crafted features-based approaches and deep learning-based approaches. The approaches that are based on hand-crafted features focus on extracting distinctive features from received signals using expert knowledge, such as I/Q quadrature modulation defects [6], statistics regarding time–frequency energy distribution [7,8,9,10], and complexity measurement [11,12]. These approaches require significant prior knowledge and achieve limited accuracy. In response, deep learning-based approaches, which learn to identify devices from data, have been extensively studied recently and have shown prominent advantages [13,14,15,16,17,18]. While most existing work has focused on classification among a closed set of known devices, the approaches that have been developed are impractical in the prevention of identity proofing since they may simply identify an unseen device as the most similar authorized device. Open-set authentication is more feasible in this scenario as it not only identifies known devices, but also rejects unseen transmitters. Although approaches for the open-set authentication of IoT devices have been proposed in the last two years [19,20,21,22,23], their performances still remain to be improved when the number of authorized devices for training is limited.

In this paper, we propose a novel deep learning framework for the accurate open-set authentication of IoT using limited authorized devices. The framework leverages the strengths of additive angular margin softmax (AAMSoftmax) [24], which promotes the discriminability of features that are learned by neural networks, and OpenMAX [25], which can effectively reject unknown classes. The codes are available at https://github.com/huangkeju/AAMSoftmaxOpenMax (accessed on 19 February 2022). The contributions of this paper are summarized as follows:We propose to adopt AAMSoftmax to enhance the discriminability of features that are learned by neural networks, so that the features of unseen devices are distributed away from those of authorized devices;We propose a modified OpenMAX method, namely adaptive class-wise OpenMAS, so that it can be combined with AAMSoftmax and unseen IoT devices can be distinguished adaptively based on the features that are learned by neural networks;We propose a framework that leverages the strengths of AAMSoftmax and OpenMAX for the open-set authentication of IoT devices. The evaluations of both simulated data and real ADS–B data show that the proposed framework was advantageous for open-set IoT authentication, especially when the number of devices for training was limited.

The remainder of this paper is organized as follows. Works related to open-set IoT authentication are discussed in Section 2. The problem is formulated in Section 3 and the proposed framework is presented in Section 4. Section 5 presents the performance evaluation and Section 6 concludes the paper. The current literature comparison can be found in Table 1.

## 2. Related Works

### 2.1. Background

The open-set authentication of IoT devices has not been investigated until recently. Generally, the open-set authentication of IoT devices is achieved using a feature extractor and a classifier (Figure 2). The feature extractor is a neural network that outputs the features of the input signals. The feature extractor network is trained to learn features that are discriminative for different devices. The classifier builds boundaries for the learned features of each authorized device at the training stage, so that it can predict whether a sample belongs to an authorized device.

### 2.2. Feature Extractor

In [19], a generative adversarial network (GAN) was utilized to identify rogue transmitters. However, training a GAN is not trivial and requires a large amount of data. In [23], hypersphere projection was used to learn more separable features in angular space, which was inspired by methods used in face recognition technology [26]. Similarly, the zero-bias layer proposed by [20] also projects features to hypersphere, but with an additional dense layer for faster convergence. In this paper, we propose to learn features not only in angular space, but also with an additive margin, which could further enhance the discirminability of features. The loss function that we adopted, i.e., AAMSoftmax, was first proposed in face recognition technology [24] and has also been applied in speaker recognition technology [27,28]. Moreover, neither [20,23] considered the classifier design for open-set authentication and simply distinguished unseen devices based on distance from authorized devices. However, the distance threshold to reject unseen transmitters can be hard to determine and may influence the final performance. Therefore, we propose to employ a modified OpenMAX classifier that can work with AAMSoftmax and adaptively distinguish unseen devices.

### 2.3. Open-Set Classifier

In [21], a novel approach was proposed for the outlier detection of Wi-Fi devices and ADS–B signals. A classifying neural network predicts slices of a packet and the statistics of those predictions are then compared to a threshold to determine whether the packet is from an unknown device. Compared to the approach that is based on this threshold, the OpenMAX employed in this paper used an activation vector to incorporate more information, which led to an improved performance [25]. In [22], five different open-set recognition approaches, namely a discriminator (Disc), discriminating classifier (DClass), one-vs.-all classifier (OvA), OpenMAX, and an autoencoder, were compared thoroughly using a Wi-Fi dataset. As Disc and DClass rely on known outlier sets for training, which require more data from additional emitters in practice, they were not considered in our evaluation. Autoencoders were also not considered because they do not make use of labels and usually lead to a moderate performance. Although OvA achieved a better performance than OpenMAX in the experiments of [22], we found that OvA fails when the unseen device is much more similar to one of the authorized devices than the others, which is common when training with limited authorized devices. By contrast, OpenMAX is less sensitive to the number of authorized devices and works more preferably in this setting, which is discussed in detail in Section 4. Furthermore, both [21,22] simply used softmax to train the neural network and did not consider ways to obtain more discriminative features in order to achieve a better performance.

## 3. Problem Definition

We considered a finite set of devices for authentication given by $A=\{A_{1},A_{2},$…$,A_{K}\}$, where *K* denotes the total number of devices. A subset of A, denoted as $A_{T}$, represents the authorized devices. Without loss of generality, we assumed $A_{T}=\{A_{1},A_{2},$…$,A_{K_{T}}\}$, where $K_{T}$ denotes the number of devices for authorization and $K_{T}<K$. The dataset for training could be defined as $D_{T}=\left\{{\left(x_{n},y_{n}\right)}_{n=1}^{N}\right\}$, where *N* denotes the number of training samples. $x_{n}$ represents the *n*th sample of the dataset, which contained *L* complex sampling points, i.e., $x_{n}\in C^{L}$. $y_{n}$ is the corresponding identity of sample $x_{n}$ and $y_{n}\in \{1,2,$…$,K_{T}\}$. A neural network with parameters Ω was trained using $D_{T}$ for open-set authentication. The neural network, which could be viewed as a mapping function $F_{\Omega }:C^{L}\to R^{K_{T}+1}$, mapped the signal x from any device of A to its prediction score p. The *i*th element of p, p(i), indicates the probability that x is from $A_{i}$ of authorized devices $A_{T}$ when $i<=K_{T}$ and represents the probability that x is from an unauthorized device when $i=K_{T}+1$.

## 4. Proposed Framework

The neural network $F_{\Omega }$ could be viewed as the combination of a feature extractor $G_{\phi }$ and a classifier $C_{\psi }$, i.e., $F_{\Omega }\left(x\right)=C_{\psi }\left(G_{\phi }\left(x\right)\right)$, where ϕ and ψ denote the parameters of the feature extractor and the classifier, respectively. The feature extractor $G_{\phi }:C^{L}\to R^{M}$ transformed signal x into a feature vector of dimension *M*, while the classifier $C_{\psi }:R^{M}\to R^{K_{T}+1}$ mapped the feature vector to the corresponding prediction score p. In this paper, we propose to utilize AAMSoftmax to train the feature extractor and use a modified version of OpenMAX as the classifier. The whole training process was divided into two stages (Figure 3). In the first stage, the feature extractor was trained using the AAMSoftmax loss function without the classifier. In the second stage, the OpenMAX classifier was trained using the fixed feature extractor.

### 4.1. Feature Extractor with AAMSoftmax

Typically, neural networks for classification are trained by minimizing the cross entropy between the prediction score and the true label:(1)$$L=-\frac{1}{N}\sum_{n=1}^{N}\sum_{i=1}^{K_{T}}I\left(i=y_{n}\right)\ln p_{n}\left[i\right],$$
where I is the indicator function, $p_{n}$ is the prediction score of sample $x_{n}$, and $f_{n}$, i.e., $f_{n}=G_{\phi }\left(x_{n}\right)$ denotes the feature vector of $x_{n}$ extracted by $G_{\phi }$. A vanilla approach to obtaining the prediction score based on f is to use a dense layer followed by the softmax function:(2)$$p_{n}\left[i\right]=\frac{e^{w_{i}^{⊤}f_{n}}}{\sum_{j=1}^{K_{T}}e^{w_{j}^{⊤}f_{n}}},$$
where $W=[w_{1},w_{2},$…$,w_{K_{T}}]\in R^{K_{T}\times M}$ are the weights of the dense layer. Combining Equations (1) and (2), the loss function with softmax becomes:(3)$$L_{Softmax}=-\frac{1}{N}\sum_{n=1}^{N}\ln\frac{e^{w_{y_{n}}^{⊤}f_{n}}}{\sum_{j=1}^{K_{T}}e^{w_{j}^{⊤}f_{n}}}.$$

However, it has been demonstrated that the neural network can easily minimize the softmax loss function by manipulating the norms of the weights and feature vectors [20,23]. More specifically, the neural network may pay more attention to easily classified samples and classes by increasing their norms and neglect samples and classes that are hard to discriminate by decreasing their corresponding norms. Therefore, the feature vectors trained by this approach are not necessarily well separated, especially for devices with similar characteristics, and the performance of open-set authentication may be limited. To address this issue, an approach that was inspired by research in face recognition technology learns features in angular space, i.e., calculates the cross entropy loss after hypersphere projection [23]:(4)$$L_{A}=-\frac{1}{N}\sum_{n=1}^{N}\ln\frac{e^{w_{y_{n}}^{\prime ⊤}f_{n}^{\prime }}}{\sum_{j=1}^{K_{T}}e^{w_{j}^{\prime ⊤}f_{n}^{\prime }}},$$
where
(5)$$w_{i}^{\prime }=\frac{w_{i}}{∥w_{i}∥},$$
(6)$$f_{n}^{\prime }=s\frac{f_{n}}{∥f_{n}∥},$$
where $∥\cdot ∥$ denotes L2 norm and s>0 is a hyperparameter that determines the radius of the hypersphere. A similar approach was proposed by [20], with addition of one dense layer in the feature extractor for faster convergence.

Although learning features in angular space helps to improve performance, this approach only requires the features of different devices to be separable and does not enforce the features of the same device to be compact. As a consequence, the features of samples from authorized devices may occupy most of the angular space, leaving limited space for unseen devices. In this paper, we propose to utilize AAMSoftmax [24], a more advanced method used in face recognition technology, to learn features in angular space with an additive margin, so that learned features are more compact for the same device and more discriminative for different devices. To introduce AAMSoftmax, we first rewrote the loss function $L_{A}$ as:(7)$$L_{A}=-\frac{1}{N}\sum_{n=1}^{N}\ln\frac{e^{s\cos\theta_{n}\left[y_{n}\right]}}{\sum_{j=1}^{K_{T}}e^{s\cos\theta_{n}\left[j\right]}},$$
where $\theta_{n}$ denotes the angle vector of $f_{n}$, with $\theta_{n}\left[j\right]$ denoting the angle between $f_{n}$ and $w_{j}$. The AAMSoftmax loss function was obtained by adding a margin to the angles:(8)$$L_{AAM}=-\frac{1}{N}\sum_{n=1}^{N}\ln\frac{e^{s(\cos\left(\theta_{n}\left[y_{n}\right]+m\right))}}{e^{s(\cos\left(\theta_{n}\left[y_{n}\right]+m\right))}+\sum_{j=1,j\neq y_{n}}^{K_{T}}e^{s\cos\theta_{n}\left[j\right]}},$$
where m>0 is a hyperparameter that determines the additive angular margin. As *m* is equivalent to the geodesic distance margin penalty in the normalized hypersphere, it can enforce an evident gap between different classes. Generally, a larger *m* value leads to more separable features, but may also make the network more difficult to converge.

AAMSoftmax can be minimized using a stochastic gradient descent algorithm. This approach provides a simple way to further improve the discriminability of features with negligible computational overheads. After training with AAMSotmax, the learned features are supposed to be more compact for the same class and more discriminative for different classes.

### 4.2. Classifier of Adaptive Class-Wise OpenMAX

With the discriminative features learned by $G_{\phi }$, another issue is to classify the signals of authorized devices and accurately distinguish the signals of unseen devices using only the training signals from authorized devices. Two approaches, namely one-vs.-all (OvA) and OpenMAX, are typically used to achieve this goal [22].

OvA trains one binary classifier network for each authorized device. More specifically, for the authorized device $A_{i}\in A_{T}$, the binary classifier $C_{i}$ is trained to predict the probability of the signal transmitted by $A_{i}$, with other authorized devices considered as outliers. During inference, signals are predicted as from $A_{i}$ when $C_{i}$ outputs the highest probability and the probability is higher than the threshold δ. Otherwise, signals are predicted as from unseen devices when all classifiers output probabilities lower than δ. Although OvA achieves a better performance than OpenMAX in [22], we claim that the performance of OvA can degrade when the characteristics of the unseen device are much more similar to one certain authorized device than others. Unfortunately, this circumstance is unexceptional, especially when the number of authorized devices is limited, as demonstrated in Figure 4.

Contrary to OvA, OpenMAX distinguishes unseen devices by modeling the distributions of the outliers of each authorized device using extreme value theory (EVT), thus OpenMAX is not sensitive to the number of authorized devices. Although OpenMAX was proposed for networks trained with softmax, we adjusted OpenMAX for networks trained by AAMSoftmax with minimal changes. Furthermore, as the original OpenMAX uses the same tail size for all classes, which may lead to performance degradation, we adaptively chose different tail sizes for each class and named the modified approach as adaptive class-wise OpenMAX.

The procedure for training adaptive class-wise OpenMAX is summarized in Algorithm 1. To model the distributions of the tail samples, OpenMAX first computed the feature centers of each class:(9)$$\mu_{k}=\frac{\sum_{n=1}^{N}I\left(y_{n}=k\right)f_{n}}{\sum_{n=1}^{N}I\left(y_{n}=k\right)},$$
where $\mu_{k}$ denotes the center of an authorized device $A_{k}$, k=1,2,…$,K_{T}$. Then, we calculated the distance distributions of each authorized device and denoted the set of features belonging to device $A_{k}$ as $S_{k}$, i.e., $S_{k}=\left\{f_{n}|y_{n}=k\right\}$. As the feature extractor $G_{\phi }$ was trained in angular space using AAMSoftmax, the distance was measured based on cosine similarity:
**Algorithm 1** The training algorithm for adaptive class-wise OpenMAX. **Input:** Set of extracted features and corresponding labels $\left\{{\left(f_{n},y_{n}\right)}_{n=1}^{N}\right\}$, with $K_{T}$ classes. **Output:** mean feature vector of each class $\mu_{k}$, Weibull model of each class $\rho_{k}$. 1: **for**
k=1
**to**
$K_{T}$
**do** 2:  Compute mean vector of class *k*, $\mu_{k}$; 3:  Find features belonging to class *k*,     $S_{k}=\left\{f_{n}|y_{n}=k\right\}$; 4:  Fit Weibull model of class *k* with adaptively chosen tail size $\gamma_{k}$,     $\rho_{k}=fit\left(\left\{d_{C}\left(f,\mu_{k}\right)|f\in S_{k}\right\},\gamma_{k}\right)$; 5: **end for** 6: **Return** means $\mu_{k}$ and models $\rho_{k}$
(10)$$d_{C}\left(f,\mu \right)=\frac{1}{2}\left(1-\cos\left(f,\mu \right)\right),$$
(11)$$\cos\left(f,\mu \right)=\frac{f^{⊤}\mu }{∥f∥∥\mu ∥}.$$

Basically, a larger distance implied that the sample was far from its respective center. The distance distribution of $A_{k}$, i.e., $\left\{d_{C}\left(f,\mu_{k}\right)|f\in S_{k}\right\}$, was used to fit the corresponding Weibull model, which was then employed to predict the probability of an outlier.

Note that we adaptively chose the tail size for each authorized device instead of pre-defining one tail size for all devices, as proposed in the original OpenMAX. The tail size defines the number of the largest distances for Weibull model fitting. A small tail size may lead to an inaccurate model, while a large tail size may increase the probability of misidentifying signals from authorized devices as unseen signals. Therefore, we adaptively chose the tail size by finding the largest tail size for each authorized device that the fitted model correctly and identified 99% of the training signals.

The inference procedure of adaptive class-wise OpenMAX is listed in Algorithm 2. We denoted the feature of the test sample x as f, i.e., $f=G_{\phi }\left(x\right)$. Different from the original OpenMAX, where the closed-set prediction score can be obtained by softmax on the activation vector, we computed the closed-set prediction score q by using softmax on the scaled cosine similarity between the feature f and each class center $\mu_{k}$:(12)$$q_{k}=\frac{e^{s\cos(f,\mu_{k})}}{\sum_{k=1}^{K_{T}}e^{s\cos(f,\mu_{k})}},k=1,2,\ldots ,K_{T}.$$

The closed-set prediction score q was then revised to obtain the open-set prediction score $\hat{q}$. Another adjustment to our algorithm was that we fixed the the number of top classes to revise in the original OpenMAX, hyperparameter α, as 1 for clarity. The results of the pre-experiment also proved that setting α=1 led to the desirable performance. With α=1, we only needed to consider the class that f was closest to, i.e., $\hat{k}=argmax_{k}\left(q_{k}\right)$. The trained Weibull model of $A_{\hat{k}}$ produced the probability of f being an outlier of $A_{\hat{k}}$ based on the distance between f and its center: $\omega =\rho_{\hat{k}}\left(d\left(f,\mu_{\hat{k}}\right)\right)$. Then, ${\hat{q}}_{\hat{k}}$ and ${\hat{q}}_{K_{T}+1}$ were revised according to $q_{\hat{k}}$ and ω, with other elements unchanged and the guarantee that $\sum_{k=1}^{K_{T}+1}{\hat{q}}_{k}=1$. Finally, predictions for open-set authentication could be made with $\hat{q}$.
**Algorithm 2** The inference algorithm for adaptive class-wise OpenMAX. **Input:** feature f of the test sample. **Require:** mean feature vector of each class $\mu_{k}$, Weibull model of each class $\rho_{k}$. **Output:**
$\hat{q}$, the prediction score of the test sample. 1: Compute the closed-set prediction score q; 2: Let $\hat{k}=argmax_{k}\left(q_{k}\right)$, compute the probability the test sample is an outlier of class $\hat{k}$, $\omega =\rho_{\hat{k}}\left(d_{C}\left(f,\mu_{\hat{k}}\right)\right)$; 3: Revise the prediction score as $\hat{q}$:    ${\hat{q}}_{k}=q_{k}$, for k=1 to $K_{T}$ and $k\neq \hat{k}$,    ${\hat{q}}_{\hat{k}}=\left(1-\omega \right)q_{\hat{k}}$,    ${\hat{q}}_{K_{T}+1}=\omega q_{\hat{k}}$; 4: **Return**
$\hat{q}$

## 5. Performance Evaluation

The proposed framework was evaluated using both simulated data and real Automatic Dependent Surveillance–Broadcast (ADS–B) data. The use of AAMSoftmax for training the feature extractor was compared to the use of the conventional softmax and ASoftmax [23], while the OpenMAX classifier was compared to an OvA classifier.

### 5.1. Evaluation Dataset

#### 5.1.1. Simulated Dataset

The simulated dataset was used as a toy problem. We generated a dataset of eight devices, denoted as $A^{S}=\{A_{1}^{S},A_{2}^{S},$…$,A_{8}^{S}\}$. The signals of each device were generated with unique pairs of I/Q gain imbalance parameter *G* and carrier leakage parameter ξ to simulate physical imperfections [29], as listed in Table 2. Although I/Q gain imbalance and carrier leakage do not cover all of the factors that can lead to emitter impairments, the generated dataset could reflect the properties of different approaches to some extent. The signals were modulated with quadrature phase shift keying (QPSK) with a symbol rate of 64 k symbols/sec, sampled at a frequency of 1024 kHz. We generated 4000 samples for each device, with each sample containing 1024 sampling points, i.e., 64 symbols. We used 3000 samples from each device for training and the remaining 1000 samples from each device for testing.

#### 5.1.2. ADS–B Dataset

The real-word ADS–B dataset was provided by [30] and included 107 devices, i.e., $A^{R}=\{A_{1}^{R},A_{2}^{R},$…$,A_{107}^{R}\}$. There were roughly 400 samples for each device. Each signal was transformed into a 32 by 32 by 3 tensor by the preprocessing approach of [30]. We used 60% of the dataset for training and the remaining 40% for testing.

### 5.2. Evaluation Using the Simulated Dataset

The backbone of ResNet [31] was employed as the feature extractor for the simulated dataset, which was composed of nine convolutional layers, as shown in Figure 5. The feature extractor took a sample of 1024 I/Q sampling points as input and outputted a feature vector of dimension 512. The networks were trained by softmax and AAMSoftmax, both at learning rate of $10^{-4}$ for 100 epochs. The batch size for training was set as 100. The hyperparameters of AAMSoftmax were set as s=10 and m=1.0. Then, both feature extractors were used to train an OvA and the OpenMAX classifiers, resulting in four combinations, namely softmax + OvA, AAMSoftmax + OvA, softmax + OpenMAX, and AAMSoftmax + OpenMAX. The threshold δ of the OvA was set as 0.99 in all combinations.

#### 5.2.1. Comparison of Overall Performance

For each combination, we successively chose one device in $A^{S}$ to be the unseen device and the remaining seven devices to be the authorized devices. The macro averaged $F_{1}$ scores [32] for the different combinations are shown in Table 3. Basically, using the OvA classifier led to a mediocre performance, regardless of whether the feature extractor was trained by softmax or AAMSoftmax, which could have been caused by defects in the OvA classifier. When OpenMAX was combined with softmax, the desirable performance was achieved when $A_{3}^{S}$, $A_{4}^{S}$, $A_{7}^{S}$ or $A_{8}^{S}$ was chosen as the unseen device. However, the combination of OpenMAX and AAMSoftmax showed a superior performance regardless of the choice of unseen device. One reason for this performance gap could be that the parameters of $A_{5}^{S}$ and $A_{6}^{S}$ were close to $A_{1}^{S}$ and $A_{2}^{S}$, respectively, making them hard to separate when trained by softmax.

#### 5.2.2. Comparison of Feature Extractors

We compared the predictions of the OvA and OpenMAX classifiers based on the features learned by AAMSoftmax and with $A_{3}^{S}$ as the unseen device. The features obtained by the AAMSoftmax training were visualized using t-SNE [33], with different colors corresponding to the true identities and predictions of different devices, as shown in Figure 6. It was shown that the learned features could properly separate different devices, including the unseen device. However, OvA misclassified most samples of the unseen device $A_{3}^{S}$ as $A_{7}^{S}$, the parameters of which were closest to $A_{3}^{S}$, as shown in Table 2. OpenMAX correctly distinguished most samples of the unseen device, although a few samples of other devices were misidentified. This comparison revealed that the OvA classifier was ineffective when the unseen device was more similar to one certain authorized device than others, while OpenMAX was less vulnerable to this condition.

#### 5.2.3. Comparison of Classifiers

The distributions of the feature distances of softmax and AAMSoftmax were compared to verify the discriminability of the learned features. Assuming $A_{j}^{S}$ was chosen as the unseen device and one of the authorized devices $A_{i}^{S}$ was chosen as the reference device, three sets of distances, namely intra-distances U, inter-distances I, and open distances O, were calculated as follows:(13)$$U=\{d\left(f_{n}^{S},\mu_{i}^{S}\right)|y_{n}^{S}=i\},$$
(14)$$I=\{d\left(f_{n}^{S},\mu_{i}^{S}\right)|y_{n}^{S}\neq i\;and\;y_{n}^{S}\neq j\},$$
(15)$$O=\{d\left(f_{n}^{S},\mu_{i}^{S}\right)|y_{n}^{S}=j\},$$
where U denotes the set of distances between the features of $A_{i}^{S}$ and the feature center of $A_{i}^{S}$, I denotes the set of distances between the features of other authorized devices and the feature center of $A_{i}^{S}$, and O denotes the set of distances between the features of the unseen device $A_{j}^{S}$ and the feature center of $A_{i}^{S}$. For softmax, $d\left(f,\mu \right)=d_{E}\left(f,\mu \right)=∥f-\mu ∥$, i.e., Euclidean distance, and for AAMSoftmax, $d\left(f,\mu \right)=d_{C}\left(f,\mu \right)$, i.e., cosine distance. We selected i=2 and j=6 for demonstration, since $A_{2}^{S}$ was highly similar to $A_{6}^{S}$. The densities of the distance distributions are shown in Figure 7. Not surprisingly, the intra-distances and inter-distances were distributed apart from each other, regardless of whether the feature extractor was trained by softmax or AAMSoftmax. However, when the feature extractor was trained with softmax, the distribution of open distances was highly overlapped with intra-distances, implying that the features were incapable of accurately separating $A_{2}^{S}$ samples from those of the unseen device $A_{6}^{S}$. As for features learned by AAMSoftmax, although the distribution of open distances was still close to that of the intra-distances, they were evidently more separable and demonstrated less overlapping.

#### 5.2.4. Comparison of Different Combinations

The confusion matrix of different combinations is shown in Figure 8, with $A_{6}^{S}$ used as the unseen device. The approaches using softmax were inferior at distinguishing the samples of the unseen device $A_{6}^{S}$ due to the fact that the features from the unseen device were overlapped with authorized devices, as shown previously. The combination of AAMSoftmax and OvA could also not distinguish the unseen device correctly because the learned features of the unseen device $A_{6}^{S}$ were still much closer to $A_{2}^{S}$ than to the other authorized devices and the OvA failed in this condition. Only the combination of AAMSoftmax and OpenMAX correctly identified the unseen device, at the cost of a slightly decreased performance concerning authorized devices. The results of the different combinations verified the significance of combining AAMSoftmax with OpenMAX.

### 5.3. Evaluation Using the Real ADS–B Dataset

We used the network of [20] as the feature extractor, with the addition of a dropout layer, as shown in Figure 9. The feature extractor took a sample of 3×32×32 tensor as input and outputted a feature vector of dimension 64. The network was trained by softmax, ASoftmax, and AAMSoftmax, all at learning rate of $10^{-4}$ for 100 epochs. The batch size for training was set as 128. The hyperparameter of the OvA was set as δ=0.9 and the hyperparameters of AAMSoftmax were set as s=3 and m=1.0. As the feature extractor ended with a fully connected layer, it was equivalent to the zero-bias layer approach [20] when trained by ASoftmax. Then, all of the feature extractors were used to train the OvA and OpenMAX classifiers, resulting in six combinations. For each combination, we successively selected 5, 10, 21, 32, 43, and 54 devices as the authorized devices and the remaining devices as unseen devices.

#### 5.3.1. Performance Comparison

The performance of each setting was assessed using the average of ten different runs, with a random selection of authorized devices and a random initialization of network parameters. The results are shown in Figure 10, with the bars indicating the averages of the different runs and the error bars denoting the standard deviation. Figure 10a displays the $F_{1}$ scores of the devices, including authorized and unseen devices when using different combinations. All combinations performed better with more authorized devices, implying that the network could learn more discriminative features with more authorized devices, which coincided with research in face recognition technology [34]. Although approaches using the OvA classifier achieved a comparable performance with a large number of authorized devices, they performed poorly with limited authorized devices regardless of the loss function that was used to train the feature extractor, confirming that the OvA classifier relied heavily on the number of authorized devices, as discussed previously. For approaches using the OpenMAX classifier, softmax + OpenMAX achieved a mediocre performance in all settings, while ASoftmax + OpenMAX and AAMSoftmax + OpenMAX achieved evidently superior performances, with AAMSoftmax + OpenMAX performing modestly better than ASoftmax + OpenMAX. Therefore, the discriminability of extracted features was critical for OpenMAX to perform successfully. By comparing the recall of authorized devices in Figure 10b and the recall of unseen devices in Figure 10c, it is illustrated that AAMSoftmax + OpenMAX was successful at distinguishing unknown devices while satisfactory at identifying authorized devices.

#### 5.3.2. Effects of Hyperparameters

We first evaluated the effects of δ, the hyperparameter of the OvA classifier. The $F_{1}$ scores of Softmax + OvA and AAMSoftmax + OvA with different δ values are shown in Figure 11. When δ increased form 0.1 to 0.9, the performance increased. However, when δ increased to 0.99 and further to 0.999, the performance of a limited number of authorized devices increased but the performance decreased with a large number of authorized devices. Therefore, we claim that 0.99 was the proper value of δ. Nevertheless, the performances with limited authorized devices were mediocre compared to OpenMAX classifier regardless of the value of δ, which further proved the defects of the OvA classifier.

The hyperparameters of AAMSoftmax, i.e., *s* and *m*, were also evaluated. The $F_{1}$ scores of AAMSoftmax + OpenMAX with different *s* and *m* values are shown in Figure 12. Large s/m values could weaken the penalty of feature compactness and lead to less discriminative features, while large m/s values could punish the features too much and make the network hard to converge. Fortunately, AAMSoftmax was tolerant of the hyperparamters to some extent and the desirable performance could be achieved within a certain range of *s* and *m*. As AAMSoftmax was equivalent to ASoftmax when s=1.0 and m=0.0, it was evident that AAMSoftmax was better than ASoftmax with appropriately selected *s* and *m* values.

## 6. Conclusions

In this paper, we proposed a novel framework for the open-set authentication of Internet of Things using limited authorized devices for training. Our contributions were as follows. Firstly, the AAMSoftmax loss function was utilized to train the feature extractor for more discriminative features. Secondly, an adaptive class-wise OpenMAX classifier was proposed for accurate open-set authentication based on learned features. Thirdly, we verified the superiority of the AAMSoftmax + OpenMAX approach through experiments with both simulated data and real-world ADS–B data. The experiments showed that AAMSoftmax + OpenMAX achieved a better performance than the other approaches, with an $F_{1}$ score of 0.91 for the open-set authentication of 107 airborne transponders when only 10 of them were used for training. A remaining challenge of physical layer authentication is the robustness of the learned features [4]; for example, changing wireless channel conditions can degrade authentication accuracy prominently [35]. Our future work will focus on enhancing the robustness of the features. 

## Figures and Tables

**Figure 1 sensors-22-02662-f001:**
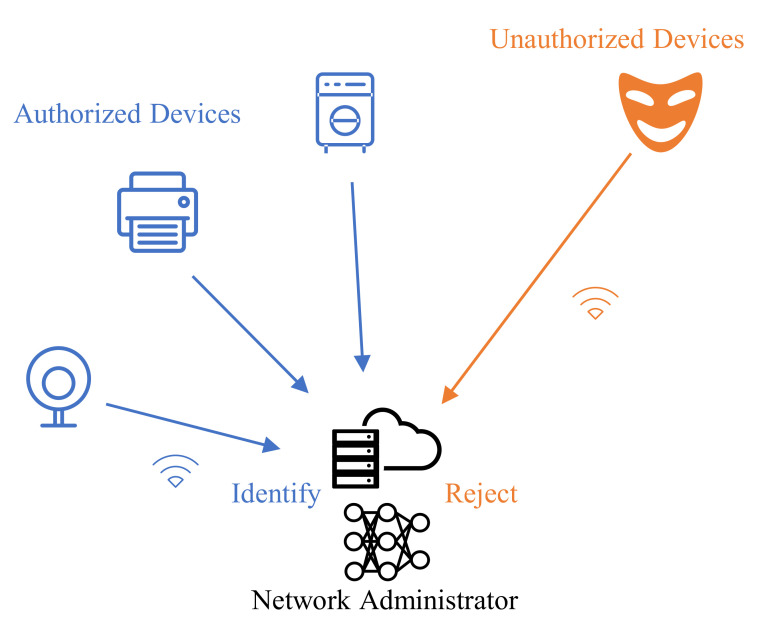
An example of device identification in the IoT.

**Figure 2 sensors-22-02662-f002:**
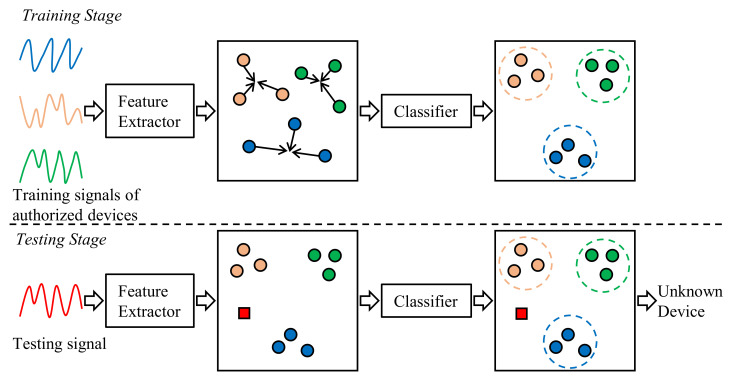
The open-set authentication of IoT devices.

**Figure 3 sensors-22-02662-f003:**
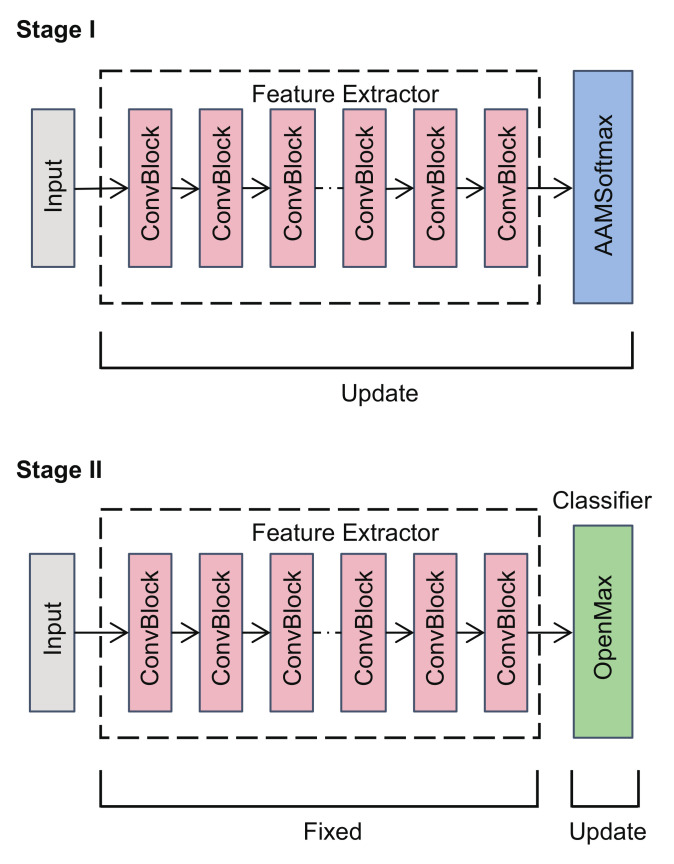
The training stages of the proposed framework. First, the feature extractor is trained by AAMSoftmax. Then, the OpenMAX classifier is updated with the fixed feature extractor.

**Figure 4 sensors-22-02662-f004:**
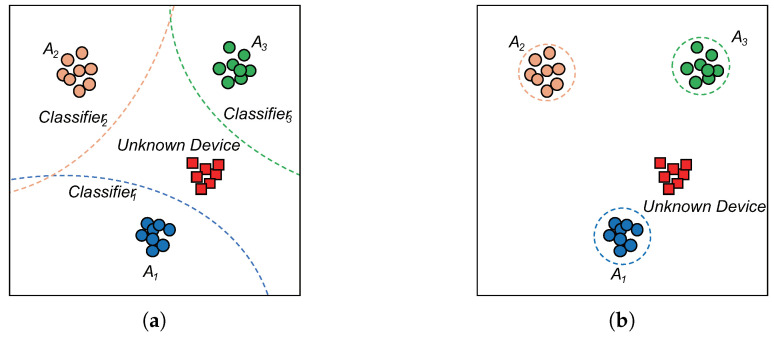
A demonstration of OvA and OpenMAX. Authorized devices are depicted as circles and unknown devices are depicted as squares. The dotted lines depict the boundaries of the classifiers. (**a**) Ova; (**b**) OpenMax.

**Figure 5 sensors-22-02662-f005:**
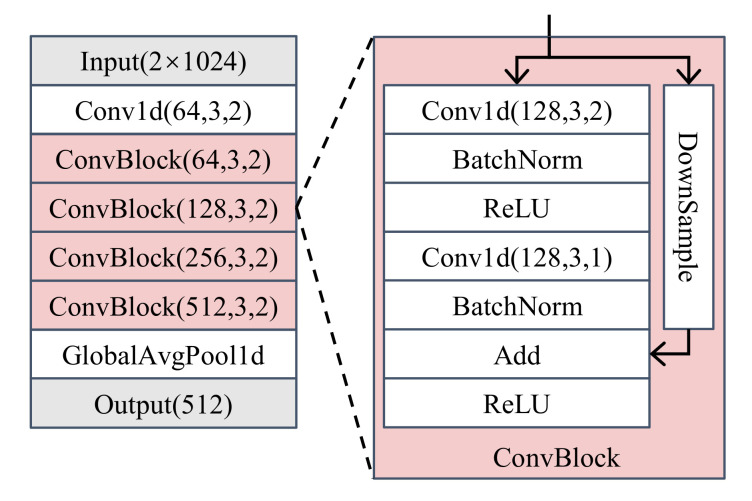
The architecture of the feature extractor for the simulated dataset.

**Figure 6 sensors-22-02662-f006:**
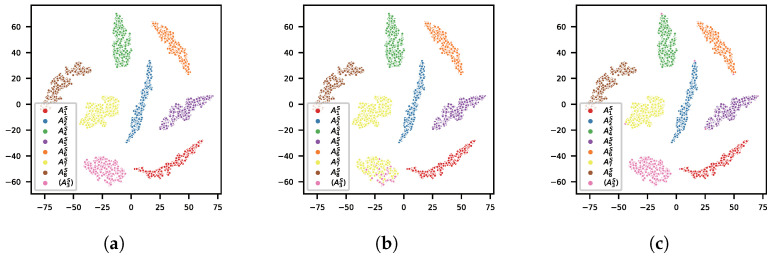
Visualizations of the true identity and predictions of the classifiers. $A_{3}^{S}$ is in brackets to denote that it is the unseen device. (**a**) True identity; (**b**) Predicitions of the OvA; (**c**) Predicitions of OpenMAX.

**Figure 7 sensors-22-02662-f007:**
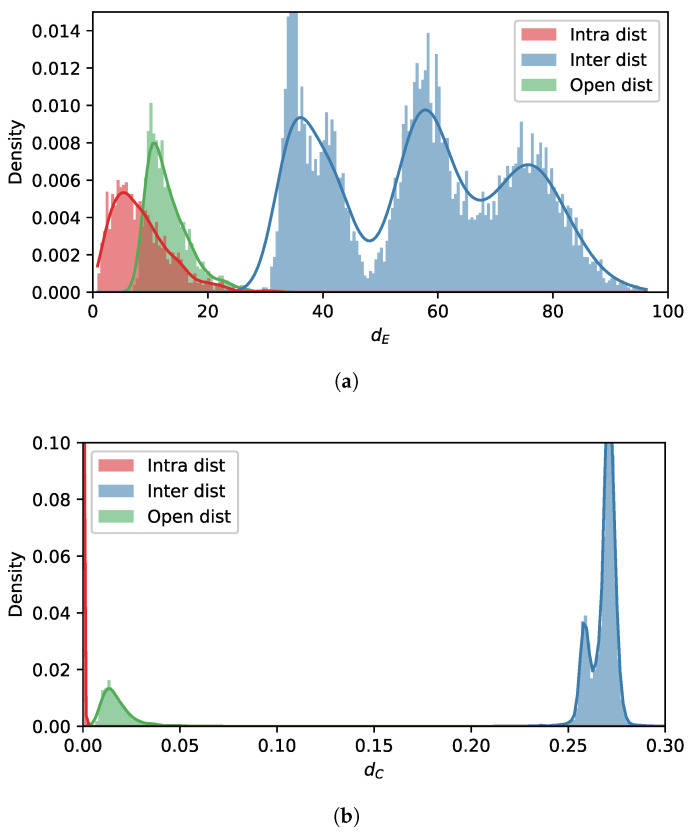
The comparison of the distance distribution of the features. ”Intra dist” indicates the distance distribution of the features from the same device with respect to the reference device, “Inter dist” indicates the distance distribution of the features from other authorized devices, and “Open dist” indicates the distance distribution of the features from the unknown device. (**a**) Softmax; (**b**) AAMSoftmax.

**Figure 8 sensors-22-02662-f008:**
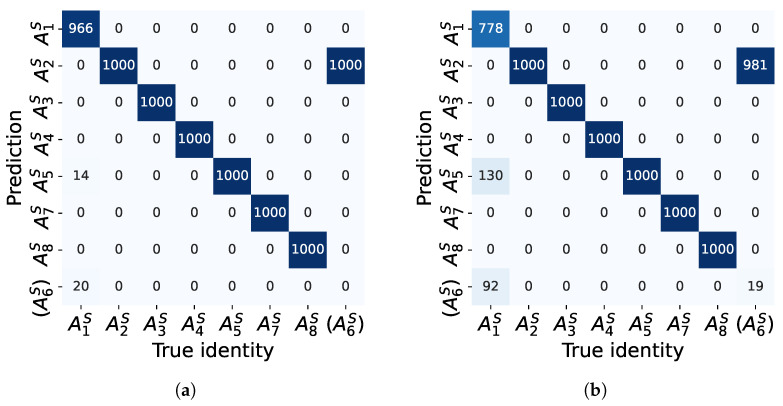
The confusion matrix of the different combinations. $A_{6}^{S}$ is in brackets to denote that it is the unseen device. (**a**) Softmax+OvA; (**b**) AAMSoftmax+OvA; (**c**) Softmax+OpenMAX; (**d**) AAMSoftmax+OpenMAX.

**Figure 9 sensors-22-02662-f009:**
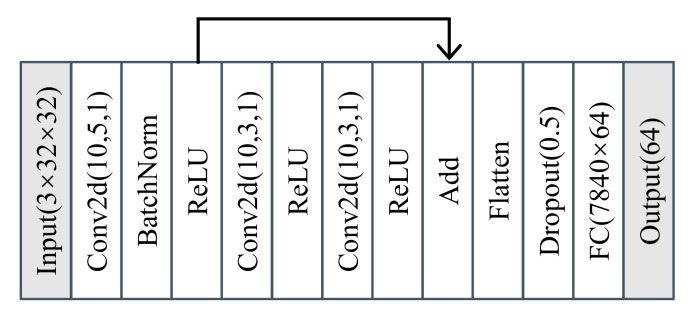
The architecture of the feature extractor for the real ADS–B dataset.

**Figure 10 sensors-22-02662-f010:**
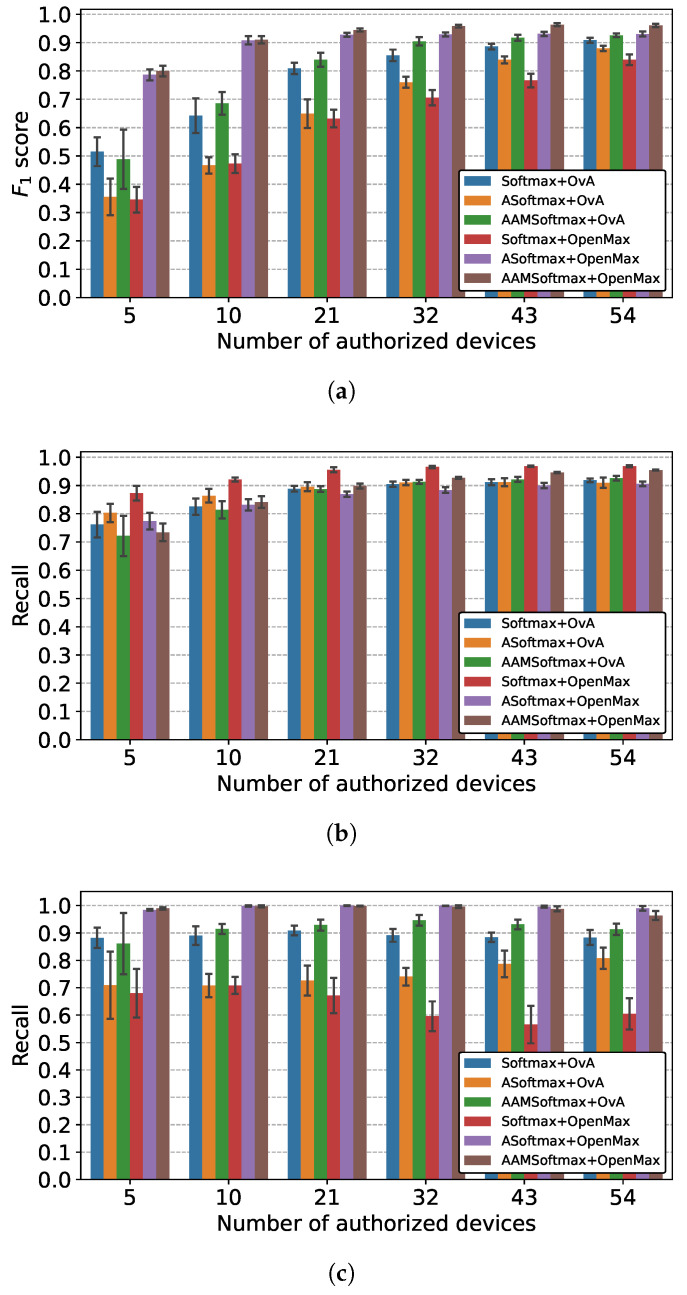
The performances of different combinations. (**a**) $F_{1}$ score of all devices; (**b**) Recall of authorized devices; (**c**) Recall of unknown devices.

**Figure 11 sensors-22-02662-f011:**
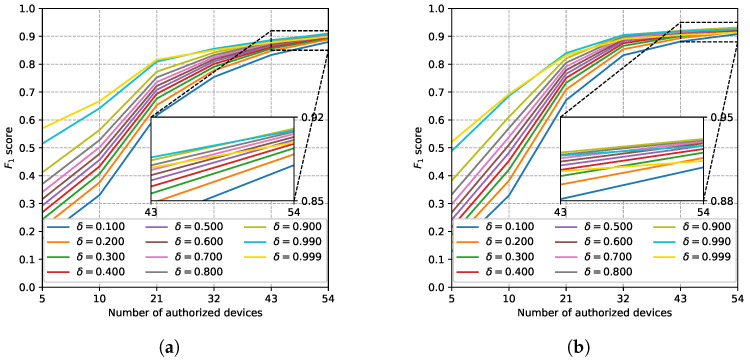
The $F_{1}$ scores of softmax + OvA and AAMSoftmax + OvA with different δ values. (**a**) Softmax+OvA; (**b**) AAMSoftmax+OvA.

**Figure 12 sensors-22-02662-f012:**
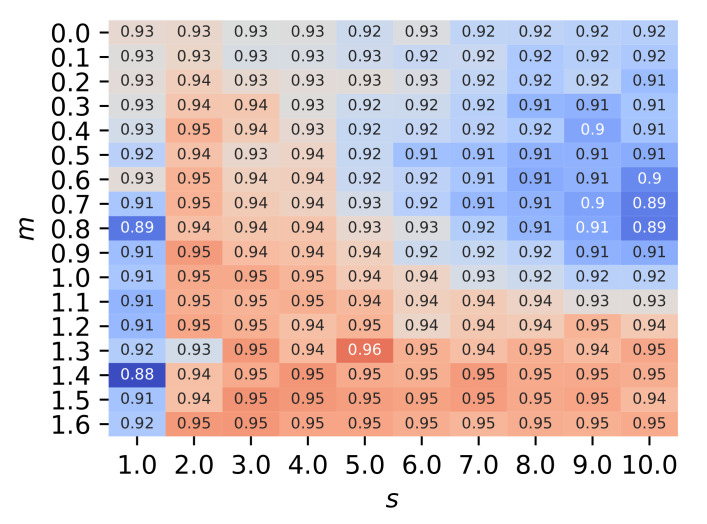
The $F_{1}$ scores of AAMSoftmax + OpenMAX when trained by 21 authorized devices with different *s* and *m* values.

**Table 1 sensors-22-02662-t001:** A comparison to the current literature.

	(Loss Function of) Feature Extractor	Classifier
[19]	GAN	GAN
[20,23]	Angular Softmax	Distance-Based
[22]	Softmax	Disc, DClass, OvA, OpenMAX, Autoencoder
Our work	Additive Angular Margin Softmax	Adaptive Class-Wise OpenMAX

**Table 2 sensors-22-02662-t002:** The parameters of the simulated devices.

Devices	$A_{1}^{S}$	$A_{2}^{S}$	$A_{3}^{S}$	$A_{4}^{S}$	$A_{5}^{S}$	$A_{6}^{S}$	$A_{7}^{S}$	$A_{8}^{S}$
*G*	0.9608	1.0408	0.9608	1.0408	0.9802	1.0202	0.9608	1.0408
$\xi (10^{-2})$	1+2j	1+2j	−2−2j	−2−2j	1+2j	1+2j	1	1

**Table 3 sensors-22-02662-t003:** The $F_{1}$ scores of different combinations.

Unknown Device	$A_{1}^{S}$	$A_{2}^{S}$	$A_{3}^{S}$	$A_{4}^{S}$	$A_{5}^{S}$	$A_{6}^{S}$	$A_{7}^{S}$	$A_{8}^{S}$
Softmax + OvA	0.8333	0.8333	0.8369	0.8552	0.7376	0.8303	0.8333	0.8331
AAMSoftmax + OvA	0.7842	0.8333	0.8438	0.9391	0.8315	0.8149	0.8326	0.8331
Softmax + OpenMAX	0.8766	0.7308	0.9628	**0.9637**	0.6010	0.8328	0.9451	**0.9741**
**AAMSoftmax + OpenMAX**	**0.9695**	**0.9629**	**0.9688**	0.9623	**0.9654**	**0.9716**	**0.9658**	0.9679

## Data Availability

Not applicable.

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
