# Peer review of "A Novel Framework for Open-Set Authentication of Internet of Things Using Limited Devices"

_sensors, 2022, doi:10.3390/s22072662_

Round 1

Reviewer 1 Report

 In this paper, the authors propose a deep learning-based framework for open-set authentication of IoT devices. The used techniques and IoT is a topic getting increasingly popular. More importantly, in the spirit of open science, the authors also released their tools to the public. Those are the positive sides of the paper. However, on the downside, the paper is subject to the following presentation flaws. As such, the author may need to revise the paper significantly before it is accepted: 

First,  it is unclear what are the settings and threat model. The idea that using machine learning to train a model to authenticate the IoT devices sounds interesting. I am not sure whether this model can be used in practice.  For example, what are the settings of the victim? What are the capabilities of the attackers (can the attacker also simulate the physical signals to go around the authentication framework?) The paper can be much stronger if the authors clearly present a threat model with the attacker capabilities and victim settings clarified. 

Second, it is unclear what is the system model. For example, the proposed framework is indeed a learning-based framework, which requires a lot of resources to be consumed to train the machine learning model. However, it is well-known that IoT devices are subject to low computational resources. In such a case,  what are the role played by each IoT device? Do we also involve them to train the model? The paper should clarify those by adding an overview of the framework. 

Third,  the authors may want to add a background section, which states the knowledge required to understand their works. While the paper is mostly accessible, it is only for experts, and it may be hard for non-experts who are working in a similar topic to understand. As such, I strongly recommend the authors add a standalone background section, presenting some necessary knowledge to improve the readability of the paper. 

Finally, it is unclear how the authors organize their evaluation. It would be great if the authors add a subsection/paragraph, which asks a few research questions that need to be answered in the evaluation section so that readers can follow your thoughts in a clear manner. 

Reviewer 2 Report

In this paper, the authors contribute a framework for open set authentication by leveraging the technology of additive angular margin softmax and OpenMax classifier. As for me, this topic is important and the flow of paper is nice, based on their research work survey, scheme design, security proofs and performance analysis in scheme.

Major improvements:

  1. Please revise the grammar error in the part of Abstract.
  2. Please add a comparisons table for the related work section and bold the idea presentation compared to the literature. It helps the readers to quickly realize the achievements in the paper by study the paper and get better visibility for the future of the paper.
  3. Some confused terms, such as One vs ALL in line 91, should give a detailed explanation.
  4. The language of the paper should be better polished. There are some strange statements.

Thus, I suggest a minor revision.

Reviewer 3 Report

The paper proposes a deep learning based framework for Open Set Authentication of IoT devices based on characteristics of the physical layer signals. The authors propose an improved methodology based on the Open Set Wireless Transmitter Authorization: Deep Learning Approaches and Dataset Considerations paper (referenced at [22])
The proposed solution is well described and supported by the results.

I think is not the scope of this paper but some references to the negative side effects of the physical layer imperfections which might influence the results, would add value and perspective to this research. 

Round 2

Reviewer 1 Report

My major concerns have been addressed. Thank the authors for their efforts.